# Absence of internal multidecadal and interdecadal oscillations in climate model simulations

Michael E. Mann[1]*, Byron A. Steinman[2] & Sonya K. Miller ⓘ [1]

For several decades the existence of interdecadal and multidecadal internal climate oscillations has been asserted by numerous studies based on analyses of historical observations, paleoclimatic data and climate model simulations. Here we use a combination of observational data and state-of-the-art forced and control climate model simulations to demonstrate the absence of consistent evidence for decadal or longer-term internal oscillatory signals that are distinguishable from climatic noise. Only variability in the interannual range associated with the El Niño/Southern Oscillation is found to be distinguishable from the noise background. A distinct (40–50 year timescale) spectral peak that appears in global surface temperature observations appears to reflect the response of the climate system to both anthropogenic and natural forcing rather than any intrinsic internal oscillation. These findings have implications both for the validity of previous studies attributing certain long-term climate trends to internal low-frequency climate cycles and for the prospect of decadal climate predictability.

[1] Department of Meteorology and Atmospheric Science, Pennsylvania State University, 514 Walker Building, University Park, PA 16802-5013, USA.
[2] Department of Earth and Environmental Sciences and Large Lakes Observatory, University of Minnesota Duluth, 2205 E 5th St, Duluth, MN 55812, USA.
*email: mann@psu.edu

It is well known that the El Niño/Southern Oscillation (ENSO) leads to interannual oscillatory behaviour in the climate, providing prospects for climate predictability at seasonal[1] and perhaps even interannual timescales[2]. Nevertheless, there remains considerable uncertainty about the nature of decadal and longer-term natural, internal climate variability. Are there preferred decadal, interdecadal, or multidecadal timescale oscillations in the climate system that are distinct from the simple red noise background of climate variability? Is there evidence that such modes are predictable at decadal and longer timescales? After decades of study, these questions still lack definitive answers.

A substantial body of research has argued for the existence of an oscillatory climate mode centred in the Pacific basin with an interdecadal (~20 year) timescale. This mode is variously referred to as the Interdecadal Pacific Oscillation (IPO)[3–6] or Pacific Decadal Oscillation (PDO)[7], arguably two manifestations of the same phenomenon. It exhibits an ENSO-like spatial pattern, but with a meridionally broader signature in the tropics, and with extratropical ocean–atmosphere interactions playing a greater role[4]. Some studies have characterised the PDO and IPO by a rather broad frequency band of decadal and interdecadal time-scales, with possibly overlapping causal mechanisms[3,8,9]. Researchers[10,11] have hypothesised that such variability might arise from coupled tropical and extratropical interactions on decadal timescales, rather than tropical or extratropical mechanisms alone, with a response that is strongly influenced by ENSO. Other analyses of both observations[12–14] and model simulations[15], however, argue for the existence of a more narrowband (i.e. true oscillatory) signal characterised by a roughly bidecadal (16–20 year) timescale that is distinguishable from the continuous noise background (a review of the substantial body of research arguing for such a signal is provided in ref. [14]).

Evidence for a multidecadal (50–70 year timescale) climate oscillation centred in the North Atlantic originated in work by Folland et al. during the 1980s[16,17]. Additional support was provided in subsequent analyses of observational climate data[18]. The confident identification of any low-frequency oscillatory climate signal, however, was hampered by the limited (roughly one century) length of the instrumental climate record and the potential contamination of putative low-frequency oscillations by forced long-term climate trends. Subsequent work in the mid-1990s attempted to address these limitations. Mann and Park[12,13] and Tourre et al.[19] used a multivariate signal detection approach, the multi-taper method singular value decomposition (MTM-SVD) method, to separate distinct long-term climate signals, while Schlesinger and Ramankutty[20] employed climate model simulations to estimate and remove the forced trend from observations. These analyses appeared to provide further evidence for a multidecadal (50–70 year) timescale signal centred in the North Atlantic with a weak projection onto hemispheric mean temperature. Mann et al.[21] presented evidence based on analyses of paleoclimate proxy data that such a signal persists several centuries back in time.

Meanwhile, analyses of one specific (GFDL) climate model by Delworth et al.[22,23] demonstrated an internal multidecadal oscillation associated with the North Atlantic Meridional Overturning Circulation (AMOC) and coupled ocean–atmosphere processes in the North Atlantic. Using the MTM-SVD signal detection approach, Delworth and Mann[24] provided further evidence for a distinct narrowband (40–60 year) multidecadal oscillation in a long (1000 year) control simulation of the GFDL model. This mode was subsequently termed the Atlantic Multidecadal Oscillation (AMO)[25] (the term was coined by M. Mann in an interview[26] with Kerr; it is sometimes alternatively referred to as Atlantic Multidecadal Variability or AMV, wherein a distinct oscillatory timescale is less clearly implied).

In most studies, the AMO surface temperature signal is found to be concentrated in the high latitudes of the North Atlantic, while the projection onto the tropics and onto Northern Hemisphere (NH) mean temperature is modest. Knight et al.[27,28] demonstrated (also using the MTM-SVD method applied to the model global surface temperature field) the existence of an AMO signal in a 1400 year control simulation of the Hadley Centre (HadCM3) coupled model with peak temperature variations approaching 0.5 °C in the high latitudes of the North Atlantic, but with a modest amplitude of only ~0.1 °C in the tropical North Atlantic, and a ~0.1 °C projection onto hemispheric mean temperature. They also identified a teleconnection of the signal into the extratropical North Pacific, implying that some of the multidecadal variability evident in North Pacific SSTs might be associated with the AMO.

Many studies have attributed the observed AMO to internal oscillatory behaviour tied to the AMOC[29–35], while others have dismissed the AMO/AMV as simply the response of North Atlantic SST to stochastic atmospheric forcing[36–38]. In addition, the AMO has been attributed largely to the response of the North Atlantic to external radiative forcing in some studies[39–43], while yet others argue that an oscillatory internal AMO signal may exist but has been misidentified due to statistical procedures that do not properly account for the forced component[26,44–46].

Clement et al.[36] coupled a slab ocean mixed-layer model (with prescribed heat transport) to atmospheric general circulation models from CMIP5 and produced spatial and temporal signatures for the AMO that are highly similar to both observations and results from simulations using fully coupled ocean–atmosphere models with interactive ocean dynamics. They therefore suggest that the AMO is the low-frequency response to high-frequency atmospheric noise. This view, however, is difficult to reconcile with apparently successful prediction experiments designed to forecast AMO-related climate fields at decadal lead times[47], results that are unlikely to be achievable for the multi-decadal component of simple red noise (by which we mean an AR(1) autocorrelated process—see Methods). Zhang[35] suggests that subpolar North Atlantic climate fields exhibit more decadal persistence and more spectral power on multidecadal timescales than can be expected for red noise. On the other hand, Mann et al.[48] show that these observations, including false apparent predictability, may be an artefact of incorrect or incomplete removal of the forced component (e.g. anthropogenic forcing and the decadal-scale recovery from major volcanic eruptions) before assessing the persistence and predictability of putative internal variability (by 'predictability', we mean predictive skill in excess of the expectations for simple red noise i.e. 'damped persistence').

Haustein et al.[43] used a two-box impulse response model to argue that most of the low-frequency variability in global and hemispheric mean surface temperature can be explained by external forcing, which implies little role for multidecadal internal variability. However, previous work[12,27] demonstrates a very weak projection of any AMO signal onto large-scale mean temperature, as the putative signal is associated primarily with a large-scale redistribution of heat, rather than a mean change in surface temperature. For this reason, past analyses[12,20,24,27] have sought to detect spatiotemporal signals in the surface temperature field itself rather than simply a signal in hemispheric or global mean temperature. While additional three-dimensional sub-surface information (e.g. measures of meridional overturning) is also useful, past studies have shown that signals associated with internal model-generated AMO-like variability are clearly detectable in surface temperature and other surface fields[22–24,27,28].

Furthermore, while methods have been proposed for removing the forced component of temperature change from historical

observations or simulations, they are often associated with significant methodological biases[44–46,49,50]. Indeed, it appears extremely difficult even in principle to objectively separate forced and internal variability components from simulations or observations that contain both[45,46,51]. Some recent studies of the AMO have thus instead analysed the CMIP5 multimodel control simulations where there is no change over time in forcing[34,36,41,52–54]. In these cases, however, spectral analyses were performed on SST means across a rigidly defined index region, a substantial limitation given that the spatial footprint of the signal is likely to vary from one model to the next.

The question of whether or not there is an internal AMO oscillation in the climate system (or for that matter, a PDO oscillation) has thus been hampered by limitations in past studies relying either on observations and/or simulations that simultaneously contain both forced and internal variability components, or control simulations that involve a single (e.g. GFDL or HadCM3) model rather than a representative multimodel ensemble or employ a restricted definition of the AMO (i.e. an SST average over some pre-defined region) that might not be flexible enough to accommodate differences in model physics across a multimodel ensemble.

In the current study, we attempt to deal with each of these past limitations. Our premise, first of all, is that if a truly oscillatory AMO or PDO signal exists, there must be a coherent large-scale pattern of variability in the climate system that may or may not cancel in a hemispheric mean, and possesses a narrow-band signature in the frequency domain that is statistically significant against the null hypothesis of coloured noise (including as a special case, the typical null hypothesis of simple red noise). Moreover, given the level of sophistication of the current generation of climate models, as indicated, for example, by their ability to capture the coupled ocean–atmosphere dynamics underlying the interannual oscillatory ENSO phenomenon, we should expect evidence for the signal across a suite of state-of-the-art climate model simulations. We must, however, allow for the possibility that the precise spatial patterns and timescales of the signal might vary from model to model, while recognising the limitations that still exist in some models when it comes to the representation of ocean–atmosphere processes relevant to decadal internal variability[55–57]. An additional caveat is that oscillatory interdecadal signals could be intermittent, excited episodically by stochastic forcing[27,58] but otherwise quiescent over time intervals lasting a century or more. Control simulations should ideally therefore span ~150 years or longer for more confident inferences regarding the nature of such variability.

## Results

**Methodological decisions.** In order to assess whether such evidence exists, we have applied, to global surface temperature fields (following seminal past studies[12,27]) in both observations and models, a signal detection tool (MTM-SVD) that is almost uniquely suited to this task. The MTM-SVD method (see Methods section for details) was introduced by Mann and Park[12] for the problem of detecting and characterising narrowband signals in spatiotemporal geophysical datasets[14,59]. MTM-SVD has been employed in more than 50 peer-reviewed studies over the past 25 years, to analyse everything from paleoclimate proxy records, to observational surface temperatures, sea level pressure and drought variables, to climate model-simulated surface, atmospheric and oceanic sub-surface fields. It has also been applied in other fields, such as wireless communication and network design (see Methods for representative references). Assessments by digital signal processing experts[60] have concluded that the method offers improved performance in detecting

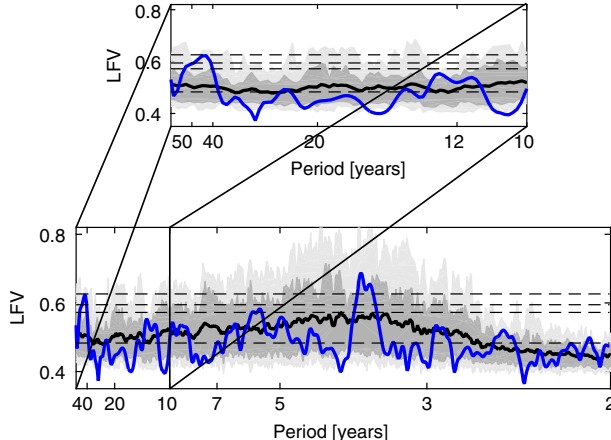

**Fig. 1 Spectra for the global surface temperature fields from control CMIP5 simulations and historical observations.** Shading with mean over all model simulations is shown by black curve and historical result is shown by blue curve. Dark grey region bounds 68% of the simulations while light grey region bounds 95% of the simulations. Lower ($f = 0.015$ cycle/year) and upper ($f = 0.5$ cycle/year) bound on frequencies shown correspond to edge of secular band and Nyquist sampling frequency. Inset zooms in on the decadal ($f = 0.1$ cycle/year) and longer periodicities. Horizontal dashed lines correspond to median ($p = 0.5$) and $p = 0.1$, 0.05 and 0.01 significance levels relative to coloured noise null hypothesis. Local Fractional Variance (LFV).

narrowband spatiotemporal signals relative to competing methods, owing to its optimal frequency-domain properties.

MTM-SVD performs a spatiotemporal decomposition of data variance locally in the frequency domain, estimating whether there is a specific large-scale pattern within a narrow frequency band that describes a larger fraction of variance than would be expected for an appropriately characterised noise process (this fractional variance as a function of frequency, termed the Local Fractional Variance or simply LFV spectrum, is used as a detection variable).

We chose to analyse surface temperature fields because (a) the putative signals in question have primarily been defined and identified based on the analysis of surface temperature data and (b) historical surface temperature observations extend far enough back in time (more than 150 years) to separate a potential multidecadal oscillation from the secular timescale variations.

**Control simulations.** We first analysed the CMIP5 control simulations (Fig. 1), requiring a minimum length of 158 years so that a putative multidecadal (40–70 year) oscillation can be resolved from a secular trend ($N = 44$ simulations satisfy this requirement (Supplementary Table 1). Nearly half of the simulations (21 out of 44) are 500 years or greater in length, spanning roughly 10 putative cycles of a multidecadal ~50 year oscillation. The CMIP5 control simulations are thus more than adequate in length to identify a multidecadal oscillation if it exists. It should be noted as well that while the temporal variations of the IPO are not well represented in CMIP5 models, the spatial pattern is generally well represented[10], suggesting that the model physics distinguishes the IPO from ENSO.

Unlike the historical observations and simulations, control simulations do not include external forcing (beyond the annual and diurnal changes in radiative forcing). Thus, they provide a much simpler laboratory wherein any apparently oscillatory behaviour must arise from internal variability. There are several noteworthy features in the results. First of all, we see a well pronounced tendency for signals in the 3–7 year ENSO band.

Indeed, even the ensemble-mean spectrum, despite the tendency for cancellation of individual spectral peaks among ensemble members, breaches significance at the $p = 0.1$ level within the ENSO band.

There are no other robust spectral features found in the control simulations, including neither bidecadal PDO (15–20 year) nor multidecadal AMO (40–60 year) timescales. The ensemble upper 16% bound on the LFV spectrum remains below the $p = 0.1$ significance level over the multidecadal (40–70 year) interval, indicating that the distribution of peaks over this frequency range is essentially indistinguishable from the expectations for random noise. It is noteworthy that neither of the two models (HadGEM2 and MPI-ESM-LR) specifically argued to exhibit an AMO signal based on analysis of the CMIP5 historical ensemble[61], show evidence of a multidecadal spectral peak for the corresponding control simulations (Supplementary Fig. 1).

**Historical observations**. These results would seem to cast doubt on the existence of either an AMO or PDO-like climate signal. However, analysis of historical observations seems to provide evidence of a narrowband AMO signal. Analysis of gridded global modern surface temperature observations (see Methods) over the 169-year period 1850–2018 (Fig. 2) reveals three spectral peaks that are statistically significant relative to the null hypothesis of coloured noise. These include two peaks within the interannual 3–7 year ENSO band and a multidecadal peak centred at a 50-year period, consistent with a putative 40–60 year AMO oscillatory signal (signals with periodicities less than half the record length, i.e. 85 years in this case, can be separated from a secular trend; see Methods). In contrast with earlier studies[12,13,62], we find no evidence for a statistically significant bi-decadal (15–20 year timescale) PDO peak using the longer, more comprehensive surface temperature data now available. Both this peak and a quasidecadal ~11 year peak identified by Mann and Park[12,13] are seen to be statistically significant in an evolutionary MTM-SVD spectral analysis for the 100 year window of 1890–1990 used in that work. These peaks disappear, however, using the

considerably longer instrumental dataset of this study (Supplementary Fig. 2), calling into question their robustness.

**Historical simulations**. We next analysed the CMIP5 historical simulations (Fig. 2; see Methods), requiring a minimum length of 158 years so that a putative multidecadal (40–70 year) oscillation can be resolved from a secular trend ($N = 118$ simulations satisfy this requirement—Supplementary Table 1). Individual spectral peaks tend to cancel when averaging over the multimodel ensemble, but even in the multimodel mean LFV spectrum there is a clear interval of elevated variance within the interannual 3–7 year ENSO band where there is greatly increased incidence of statistically significant (i.e. $p < 0.05$) spectral peaks. There is an average of 2.8 statistically significant spectral peaks in the ENSO band per simulation (which compares favourably with the two spectral peaks observed in the actual observations). Even more striking, however, is the multidecadal peak centred at the same frequency ($f \sim 0.02$ cycle/year) as in the observational data. This peak is highly robust, statistically significant even in the mean (and median) over all simulations. The peak is similar in both frequency and amplitude to the corresponding peak in the historical observations.

In contrast with the previous analysis of CMIP5 control simulations, these latter analyses of the historical period seem to provide consistent evidence in both observations and climate model simulations for an AMO signal. However, the interpretation of these analyses is complicated, as discussed earlier, by the fact that both forcing (anthropogenic and natural) and internal variability contribute to the observed variability. A parallel analysis of anthropogenic-only (see Methods) historical simulations yields a spectral peak that is also robust but centred at a lower frequency (closer to 60 years) (Supplementary Fig. 3). If the AMO signal in question reflects internal variability, it is curious—and indeed worrying—that it's character would appear to depend on the types of forcing included.

It is instructive to examine the spatiotemporal characteristics of the putative AMO signal in observations and simulations more closely. In the latter case, we have chosen as examples two models —HadGEM2-ES and MPI-ESM-LR—that have previously been argued to exhibit a strong AMO signal in analyses of CMIP5 historical simulations[61]. We find in each case a spatial pattern (Fig. 3) that emphasises the North Atlantic to some degree, consistent with an AMO signature, but the pattern is also similar to the estimated response to anthropogenic sulphate aerosols[63]. While the temporal signal (Fig. 3) is suggestive of a multidecadal cyclicity, the phase of the signal is synchronised for all three cases —observations and the two model simulations—during the latter half century, with positive peaks near 1940 and 2000 and a negative peak near 1980. If the AMO is an internal oscillation, there is no reason that its phase should be synchronised among three independent realisations (the observations and the two different model simulations).

**Discussion**

The above results were compared (Fig. 3) to an AMO study by Mann et al.[26] that examined the performance of competing methods of defining the AMO from observations. Mann et al.[26] show that a commonly used approach for estimating the AMO —linear detrending followed by a low-pass filter—leaves behind residual low-frequency (anthropogenic and natural) forced variability that masquerades as an apparent internal AMO oscillation. The study showed that a more rigorous approach that uses the CMIP5 multimodel ensemble mean to estimate and then remove the forced signal yields an entirely different and lower-amplitude AMO series. They showed that the

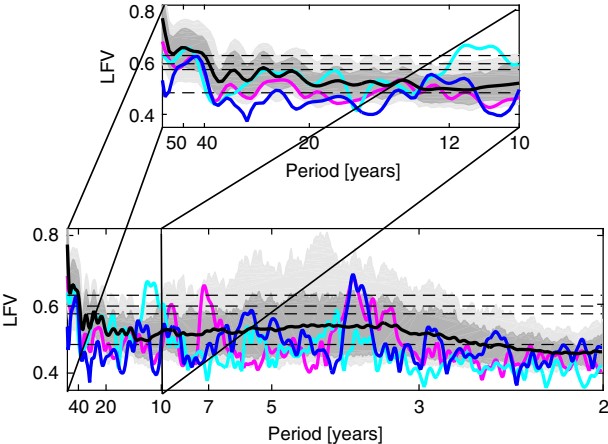

**Fig. 2 Spectra for the global surface temperature fields from historical CMIP5 simulations using natural and anthropogenic forcing and historical observations.** Shading with mean over all model simulations is shown by black curve and historical result is shown by blue curve. The HadGEM2-ES (purple) and MPI-ESM-LR (cyan) simulations are shown for comparison. Lower ($f = 0.015$ cycle/year) and upper ($f = 0.5$ cycle/year) bound on frequencies shown correspond to edge of secular band and Nyquist sampling frequency. Inset zooms in on the decadal ($f = 0.1$ cycle/year) and longer periodicities. Horizontal dashed lines correspond to median ($p = 0.5$) and $p = 0.1$, 0.05 and 0.01 significance levels relative to coloured noise null hypothesis. Local Fractional Variance (LFV).

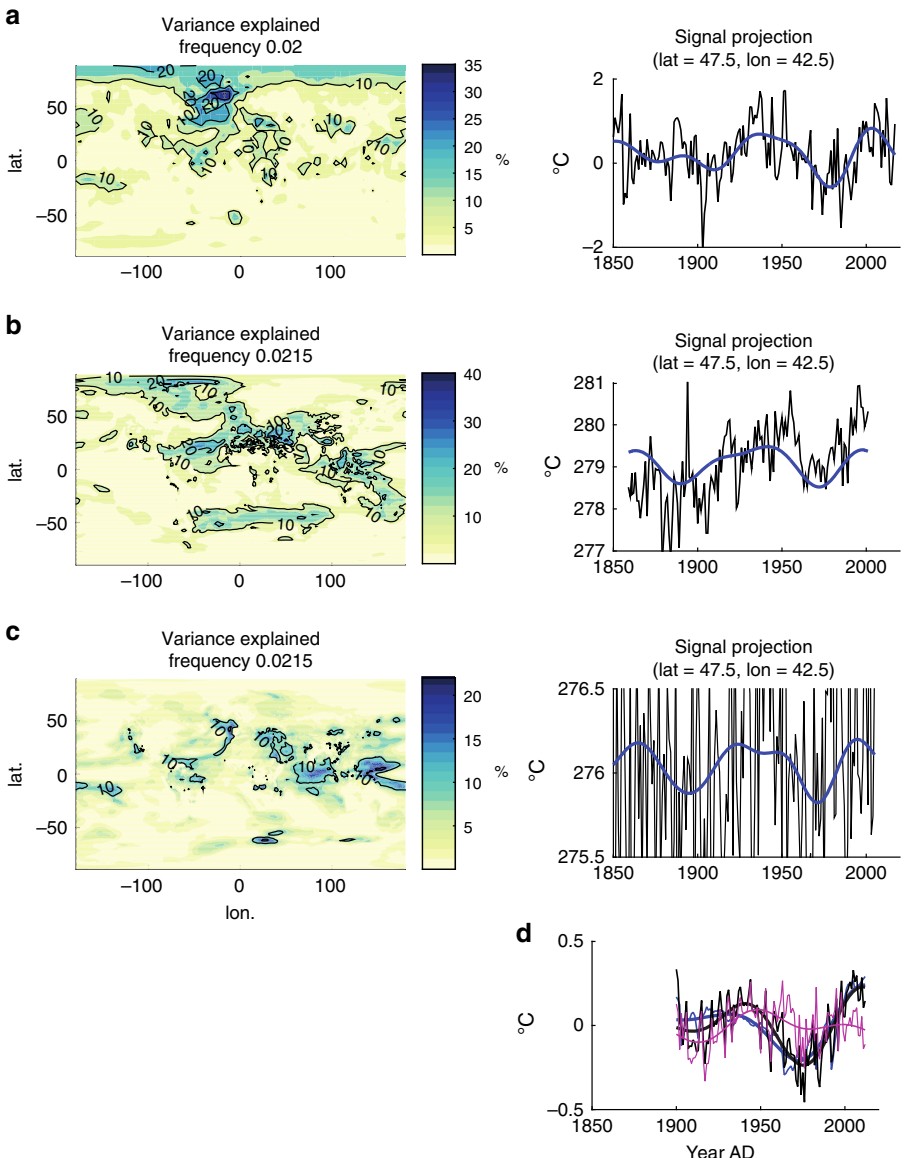

**Fig. 3 Spatial pattern of explained variance and time domain signal for AMO. a** observations, **b** HadGEM2-ES model simulation and **c** MPI-ESM-LR model simulation. Shown for comparison is the analysis by Mann et al.[26] discussed in text. In **a**–**c**, the raw temperature time series (black) and reconstructed Atlantic Multidecadal Oscillation (AMO) time-domain signal (blue) are both shown. In **d**, shown are the linearly detrended historical Northern Hemisphere (NH) mean temperature (black thin), and its 40 year smooth (black thick), the linearly detrended CMIP5 multimodel mean NH mean temperature (blue thin) and its 40 year smooth (blue thick), and an estimate of the true AMO internal series (purple thick) and its 40 year smooth (purple thick) based on removing the estimated forced component (CMIP5 multimodel mean) from the instrumental NH mean temperature. The x-axis on right side panels is year AD.

detrended AMO approach yields an inflated apparent AMO signal with precisely the features mentioned above: positive peaks near 1940 and 2000 and negative peak near 1980. Those features were shown to be largely an artifact of the substantial 1950s–1970s aerosol surface cooling trend masquerading as part of an AMO oscillation. Mann and Emanuel[44] came to a similar conclusion.

It is worth noting that some individual models do indeed exhibit an AMO-like multidecadal signal in the control simulations. Consider for example GFDL ESM-2G, which exhibits a distinct spectral peak centred at ~40 year period that is significant at the $p < 0.01$ level, with a spatiotemporal pattern that is indicative of an AMO-like signal (Fig. 4). These models are nonetheless the exception to the rule, with, as noted earlier, fewer than

7 out of 43 models breaching the $p = 0.1$ significance level (we would expect at least 4 based on chance alone).

Based on the available observational and modelling evidence, the most plausible explanation for the multidecadal peak seen in modern climate observations is that it reflects the response to a combination of natural and anthropogenic forcing during the historical era. Moreover, there is no compelling evidence from control simulations for any robust interdecadal or multidecadal climate oscillations, with the only signals that are distinct from coloured noise found within the interannual ENSO frequency band. While this does not prove that physically-based interdecadal and multidecadal modes of variability do not exist, it does call into question whether they can be classified as an oscillation (i.e. a narrowband signal).

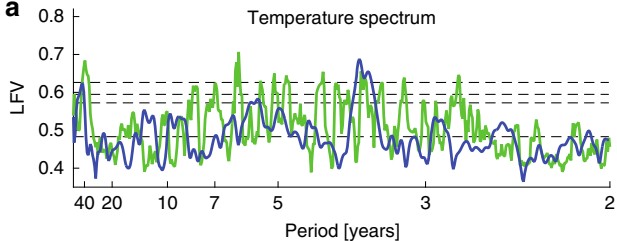

**a**

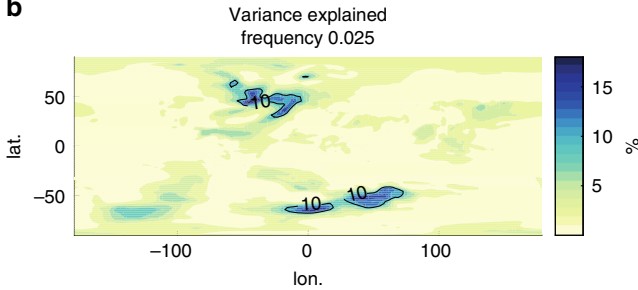

**b**

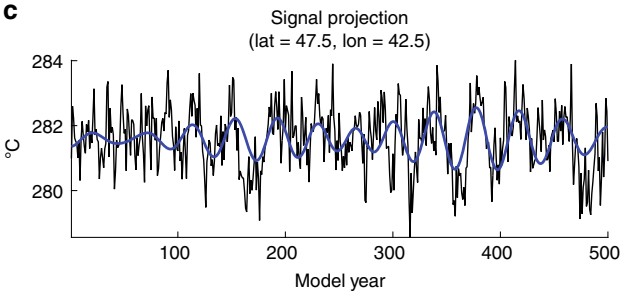

**c**

**Fig. 4 Spectra of GFDL ESM-2G model control simulation. a** Local Fractional Variance (LFV) spectrum (green; LFV spectrum for observational data is shown in blue for comparison, along with $p = 0.5, 0.1, 0.05$ and 0.01 significance levels), **b** spatial pattern of resolved variance associated with signal and **c** reconstructed time-domain signal for representative North Atlantic grid box (5° grid box centred on longitude = −42.5 and latitude = 47.5).

This finding has a number of important implications. The lack of evidence for bidecadal or multidecadal oscillations that are distinct from red noise calls into question prospects for skilful initial value decadal forecasts based on the assumption of predictable internal variability. Some recent work suggests that apparent predictability in these forecasts arises mostly or entirely from the specification of forcing (e.g. the predictable warming following large volcanic eruptions in 1982 and 1991)[48]. Our findings, moreover, call into question the past attribution to interdecadal and multidecadal climate cycles of a variety of climate trends including recent increases in North Atlantic sea surface temperatures and Atlantic hurricane activity[44]. Our findings also motivate a re-evaluation of evidence[21,24] for low-frequency climate oscillations in paleoclimate proxy data. Such apparent oscillations could reflect either internal or externally forced low-frequency climate variability. Parallel analyses of CMIP Last Millennium simulations and long-term paleoclimate proxy data, which might shed further light on this matter, constitute the subject of potential future study.

## Methods

**MTM-SVD.** The MTM-SVD methodology has been employed in more than 50 studies over the past 25 years. Applications include the analysis of global surface temperatures[12,64], precipitation[65,66], drought[67,68], coupled patterns in multiple climate fields including surface temperature, sea level pressure, winds and sea ice[13,24,64,69–85], paleoclimate proxy data[69,86] and proxy-based climate field reconstructions[24,87]. Applications in other fields include wireless communication and network design[60,88,89].

The MTM-SVD method is described in detail in the Mann and Park[14] review article, which includes an extensive discussion of the theoretical motivation and underlying assumptions, applications to synthetic examples that demonstrate the efficacy and performance of the method in detecting narrowband spatiotemporal signals embedded in red noise, and applications to observational atmospheric, oceanic, and paleoclimatic data sets. The method has also been summarised in Jolliffe[59].

A brief summary of the method is provided here.

MTM-SVD performs a singular value decomposition (SVD) of a multivariate dataset in the frequency, rather than—as in e.g. standard Principal Component Analysis (PCA)—the temporal, domain. For each time series, a decomposition is performed at each frequency $f$ over a bandwidth of $\pm p f_R$, where $f_R$, the Rayleigh frequency, is the minimum resolvable frequency, $f_R = 1/N\Delta t$, for a dataset of $N$ samples with temporal spacing $\Delta t$ and $p = K - 1$, where $K$ is the number of orthogonal (Slepian) data tapers (i.e. windowing functions) used in the Multitaper Method (MTM) of spectral analysis. Slepian tapers are mathematically equivalent to prolate spheroidal wavefunctions, and represent the solution to a variational problem which minimises spectral leakage outside a central band of width $\pm p f_R$ in frequency space. The first $K$ data tapers are optimally resistant to spectral leakage, so that the choice of $K$ in MTM represents a tradeoff between degrees of freedom (and thus, the variance in the spectrum estimate) and resolution of the spectrum. The choice $K = 3$ provides multiple (3) spectral degrees of freedom for only a modest widening (factor $p = 2$) of frequency resolution, making it a convenient choice for many applications. The first $K = 3$ data tapers can be thought of as the three lowest-order possible temporal modulations of a carrier signal of frequency $f$.

The MTM-SVD decomposition of the multivariate dataset of $M$ spatially distributed time series (which could correspond to a single spatial field like surface temperature, or a joint analysis of multiple spatial fields such as surface temperature, sea level pressure, etc.) into $K$ orthogonal modes at frequency $f$ takes the form:

$$\mathbf{A}(f) = \begin{bmatrix} w_1 Y_1^{(1)} & w_1 Y_2^{(1)} & \dots & w_1 Y_K^{(1)} \\ w_2 Y_1^{(2)} & w_2 Y_2^{(2)} & \dots & w_2 Y_K^{(2)} \\ \vdots & & & \\ w_M Y_1^{(M)} & w_M Y_2^{(M)} & \dots & w_M Y_K^{(M)} \end{bmatrix} \qquad (1)$$
$$= \sum_{k=1}^{K} \lambda_k(f) \mathbf{u_k}(\mathbf{f})^{\dagger} \mathbf{v_k}$$

where $w_i$ represents the relative weights on each of the $M$ data series (e.g. a cosine latitude factor if accounting for areal weighing of data provided on a uniform latitude x longitude grid), and $Y$ is the spectral estimate corresponding to the $l^{\text{th}}$ data taper $w_n(l)$ for the (appropriately normalised) $m^{\text{th}}$ time series $x$,

$$Y_l^{(m)}(f) = \sum_{n=1}^{N} w_n^{(l)} x_n^{(m)} e^{i2\pi fn\Delta t} \qquad (2)$$

The eigenvalues $\lambda_k$ describe the relative amplitudes of each of the $K$ orthogonal modes, while the complex left $M$-eigenvector $u_k$, the spatial Empirical Orthogonal Function (EOF), contains the spatial amplitude and phase information for that mode, and the complex right $K$-eigenvector $v_k$, the spectral EOF, describes the relative combination of the $K$ independent data tapers that characterises the temporal modulation envelope of the oscillatory signal. $W_n^{(l)}$ denotes the $l$th data taper.

The fraction $\lambda_1(f)/\Sigma \lambda_1(f)$, denoted the Local Fractional Variance (LFV) spectrum, is the detection parameter in the MTM-SVD routine. It measures the fraction of multivariate data variance locally in a bandwidth centred on frequency $f$ that can be described in terms of a single mode (i.e. a particular pattern of temporal modulation of the central carrier frequency of interest). In other words, it establishes whether there is a single spatially coherent oscillatory signal (i.e. what we define as an oscillation) centred at frequency $f$ that describes more variance than expected from noise (discussed further below).

If a spatiotemporal signal centred at frequency $f_0$ is detected, it can be reconstructed via

$$\tilde{x}_n^{(m)} = \gamma(f_0) \Re \left\{ \sigma^{(m)} u_1^{(m)} \tilde{A}_1(n\Delta t) e^{-i2\pi f_0 n\Delta t} \right\} \qquad (3)$$

where $m$ denotes spatial grid location $m$, $u_1^{(m)}$ denotes the loading of the principal spatial EOF for that grid location, $\sigma^{(m)}$ is the standard deviation of the grid box time series, $n\Delta t$ is the discrete time value (e.g. year), and $\gamma(f_0) = 2$ for signals outside the secular band (i.e. $f_0 > p f_R$), which applies for all signals of interest in our analyses.

$$\tilde{A}_1(n\Delta t) = \sum_{l=1}^{K} \xi_l^{-1} \lambda_l(f_0) \left(v_1^{(l)}\right)^* w_n^{(l)} \qquad (4)$$

is the temporal envelope modulating the sinusoidal oscillation at frequency $f_0$, constructed as a linear combination of the $K$ Slepian data tapers $w_n(l)$ weighted by the eigenvalue and the corresponding components of the principal spectral EOF $v_1^{(l)}$. The $\xi_l$ are bandwidth retention factors associated with each of the $K$ data tapers.

The typical null hypothesis for climate variability is simple red noise, which can be understood (e.g. Hasselmann[90]) as the response of a system with thermal inertia (e.g. the oceans) to high-frequency white noise (e.g. idealised weather noise) forcing. It is characterised by a single decorrelation or 'persistence' timescale, which is tied, in the case of a discrete time series, to the lag-one temporal autocorrelation coefficient of a simple AR(1) autocorrelated noise process. The MTM-SVD method invokes a null hypothesis of a coloured noise spectral background that includes, as a special case, a simple red noise spectrum. The only additional assumption is that the spectrum varies modestly in amplitude over the narrow bandwidth of the analysis (i.e. it is locally white, which is to say that the spectrum varies relatively modestly with frequency, so on the scale of the narrow spectral bandwidth of the analysis, it looks flat, i.e. white). Mann and Park[14] show that this assumption holds extremely well for the moderately red spectra encountered in climate data. It also holds, in fact, for more general noise spectra, including for example compound red noise, which is characterised by not just one but two distinct persistence timescales and has been argued to apply to some climate series[91].

One can calculate significance limits based on the above null hypothesis through a Monte Carlo approach, and peaks in the LFV spectrum of a dataset that exceed the 95% ($p = 0.05$) significance level, for example, can be interpreted as indicative of a potential oscillatory spatiotemporal signal centred at frequency $f$ in the dataset. The associated spatiotemporal signal can be reconstructed by truncating the sum in Eq. (1) at $k = 1$.

The MTM-SVD analysis was performed using $K = 3$ data tapers and a time-frequency bandwidth product of $NW = 2$. The LFV spectra and significance levels have been renormalised so they are comparable to those for the observations, despite differing spatial degrees of freedom (the observations cover a more restricted portion of the globe).

**Observational surface temperatures**. We analysed annualised global monthly average surface temperature field (Surface Air Temperature over land and Sea Surface Temperature over oceans) using the Cowtan & Way (1850–2018) surface temperature dataset[92], which infills the missing data in HadCRUT4 via kriging of the HadCRUT4 land and ocean surface temperature dataset, allowing for more direct comparisons in terms of spatial sampling with the model simulation surface temperature fields, but with additional caveats involving the interpolation schemes used. An evolutionary MTM-SVD analysis with a 100 year moving window (Supplementary Information) produces nearly indistinguishable spectral features (e.g. a statistically significant 16–18 year and 10–11 year spectral peak) to those obtained by the Mann and Park[12] earlier analysis of a considerably sparser grid consisting of only continuous temperature data for the 100 year window that corresponds to the interval (1891–1990). This confirms that similar conclusions are reached for the same time period for two very different versions (sparse grid with only nearly continuous gridbox series vs. nearly global grid based on interpolation of missing data) of the instrumental dataset.

**CMIP5 control simulations**. We analysed the global gridded surface temperature fields from the Coupled Model Intercomparison Project Phase 5 (CMIP5)[93] pre-industrial control multimodel simulations ($N = 48$ realisations; $M = 46$ models; a subset $N = 44$ realisations have the required minimum length of 158 years, as discussed in main text). These simulations used prescribed pre-industrial conditions of atmospheric concentrations or non-evolving emissions of gases, aerosols or their precursors, as well as static land use.

**CMIP5 historical simulations**. We also analysed the global gridded surface temperature fields from the CMIP5 historical experiment multimodel ensemble simulations, including both the anthropogenic + natural forced simulations ($N = 164$ realisations; $M = 48$ models) and anthropogenic-only forced simulations ($N = 40$ realisations; $M = 8$ models) spanning 1850–2005 (Supplementary Table 1). Each physics version of a model was considered a separate model.

## Data availability
All raw data and results are available at the supplementary website: http://www.meteo.psu.edu/~mann/supplements/Mann_MTMSVD_2019/Data.

## Code availability
All ©Matlab code is available at the supplementary website: http://www.meteo.psu.edu/~mann/supplements/Mann_MTMSVD_2019/Code.

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

## Acknowledgements

We acknowledge the World Climate Research Programme's Working Group on Coupled Modelling, which is responsible for CMIP, and we thank the climate modelling groups for producing and making available their model output. M.M., B.S. and S.M. were all supported by a grant from the NSF Paleoclimate Program #1748097.

## Author contributions

M.E.M. conceived, designed and performed the research and wrote the paper. B.A.S. and S.K.M. performed some aspects of the research and co-wrote the paper.

## Competing interests

The authors declare no competing interests.
