## [Peer Review File · Nature Communications]

Reviewers' comments:

Reviewer #3 (Remarks to the Author):

Response to the authors

My apologies to the authors for missing that the analysis was extended to a large multi-model ensemble rather than the two examples used as illustrations.

In balance I like the scope of this article in applying a flexible spectral method to a large number of ensemble experiments. I feel that the significance testing is sound -- that is the statistical tests for spectral peaks hold their level. Thus this article confronts the community with a large number of control climate simulations that do not detect long term periodicities. "failing to reject the null hypothesis of no long term periodicities" Perhaps this finding justifies publication in Nature.

I note that this result comes with the focus on a single method: MTM-SVD. Given that the author is the developer of this method, it is natural for him to be a proponent. However, the issue of the power (in a statistical hypothesis testing context) of the MTM-SVD method when failing to reject a null hypothesis should be considered but given the authors response I suspect this is beyond the scope of their analysis and interests. A more positive approach to understand the power is to let the publication of this work to spur more scrutiny of the MTM-SVD method's (statistical) power and comparison with other approaches.

Reviewer #4 (Remarks to the Author):

General

As a new reviewer, it is clear that this paper has caused some confusion, and doubts about whether this paper is novel enough for Nature Communications. A basic problem is that the title does not accurately represent the content of the paper. Another problem for publication in Nature Communications is that current climate models, and certainly CMIP5 models run in control mode, are not good enough to make definitive statements about the reality or otherwise of narrow band natural multidecadal and interdecadal variability. They are of relatively low resolution and it is very likely that much higher resolution is needed to capture longer time scale natural variability well. New, if preliminary, physical insights into fundamental model biases relating to resolution are in Scaife et al (2019), especially the last section of that paper. These insights are presented in the context of seasonal forecasts but have implications for all time scales and ocean-atmosphere interactions. Thus, it has subsequently been found that all current climate models suffer from the now well-known signal to noise paradox (Eade et al, 2014)) out to all time scales, also mentioned in Smith et al (2019) in the context of decadal prediction. This seems likely to especially affect the North Atlantic region and ocean-atmosphere interactions there. However, the results in this paper, with the modifications and additions suggested here, are certainly interesting enough for a more specialised publication, given acknowledgement of this potential problem but may be more marginal for Nature Communications. Having said that, if the somewhat wider perspectives suggested here were included, they would increase the suitability of the paper for Nature Communications. However, the analyses appear to need some modification as well, as mentioned below, to best make their points. Finally, the paper omits other references which cast significant light on how the topics in this paper should be approached, added below.

Having said this, there is nothing wrong as far as I can see with the statistical methods, though the paper does suffer a little from lacking an explicit dynamical background when discussing low frequency climate variability. As the authors clearly point out, it is clear that the observational record of the AMO and the PDO/IPO has been significantly affected by human-kind mainly via anthropogenic aerosols. The statistical approach of this paper may obscure this truth a little, but the quoted literature is clear, though the magnitudes of anthropogenic effects need more clarification in future. The authors in fact miss out an important reference about an anthropogenic influence on the PDO/IPO, listed below. This would strengthen the points already made about the observed record.

Specific

1. I suggest that the title becomes "Absence of internal narrow band multidecadal and interdecadal

oscillations in control simulations of the climate system”.

2. Lines 23-33. A prominent c20 year PDO/IPO oscillation may be overstated as being current thinking. There is no settled theory of the PDO/IPO physics but there are a number of candidate processes which could give rise to several overlapping PDO/IPO time scales. These are summarised in the review by Henley (2017). This complexity tends to fit the analysis by Folland et al (1999) where their EOF3, which became the IPO of Power et al (1999), shows a vague power spectrum and a complex observational time series. Henley (2017) shows many spectra which support this vagueness as well as clear differences between the many analyses. Thus for the IPO, the reality of interdecadal to multidecadal variability might not depend on identifying a specific narrow band signal.

3. Lines 34-56. This paper deliberately does not include paleoclimate analyses. However, this could be a serious weakness in the longer term for determining the truth about narrow band internal AMO oscillations. For instance, Chylek et al (2011) provide evidence for narrow band AMO oscillations around 20 and 50 years going back many centuries. If the paper, as modified, does not include paleo-analyses, this is all the more reason to be cautious about CMIP5 control model results.

4. Somewhere in the introductory section another caution should be expressed, given relatively short model or observational results. Wavelet analyses of the climate often show that specific narrow band oscillations are episodic. This is, for instance, seen in the Knight et al (2005) (quoted in the refs) 1400 year control model results and the Chylek et al (2011) paleoclimate results. So this adds an extra dimension to interpreting model or observational results. Episodes where the narrow band is missing can be a century, or more. long. This could affect ensemble member control runs 150 years long where in some members a narrow band oscillation exists and in others of the same model it does not. This would not necessarily be an error, but it would make detection of narrow band oscillations in short control runs more challenging. Of course, this might also mean that the modern short observational record could overstate the long-term frequency or persistence of a specific apparent narrow band feature.

5. Line 140-143. It is not stated which surface temperature fields are being analysed and the captions miss this information. Are these global mean fields and time series? It would be better, given that the paper concentrates on the PDO/IPO and AMO, to use fields that best pick up these oscillations. These would be the North Atlantic region and an appropriate Pacific region. The need to do this is shown by the regional oceanic SST spectra of Folland et al (1999), which clearly differ a lot.

6. Lines 176 -230. An addition paragraph is needed. Having criticised CMIP5 models, there is a paper showing that most CMIP5 models reproduce the spatial pattern of the IPO well, though they do not perform so well on its temporal variations (Henley et al (2017). This suggests that there is genuine IPO physics distinguishing the IPO from ENSO, even if CMIP5 model temporal variability is poor. Furthermore, it suggests that the models pick up some of this physics. A “real” IPO (narrow band or not) is consistent with the observational paper of Folland et al (2002) showing a distinctive influence of the IPO compared to ENSO on the South Pacific Convergence Zone.

7. Lines 188-196 (probably). Add a mention of Smith (2016) which shows that anthropogenic aerosols have likely quite recently influenced the PDO.

8. Lines 231-235. This part of the last paragraph of the main text is not sound. Not only does it overinterpret the results of this paper, but it contradicts the recent decadal prediction paper by Smith et al (2019) showing recent advances in prediction skill. Decadal prediction skill is clear over years 2-9 ahead in several variables, with additional predictability from predicted forcing changes. There clearly is skill in decadal forecasts this far ahead, particularly in the Atlantic sector. Thus Meehl et al (2014) (quoted in the refs) show generally useful CMIP5 decadal forecasting skill of the AMO out to at least years 3-6. Initial conditions themselves in decadal forecasts are of course influenced by anthropogenic effects, quite apart from skill due to forecast changes in forcing. So the control model results in this paper have few serious implications for forecasts of periods as short as a decade or somewhat less, the current time scales of decadal forecasts.

Minor comment

1. The caption to Fig 1 is wrong. It should read “control” on the first line. In Fig 1 and 2 captions, the a and b parts of the figures should be described under these letters.

References

Scaife et al., 2019. Does increased atmospheric resolution improve seasonal climate predictions?

Atmos Sci Lett., 20, doi: 10.1002/asl.922

Eade, R., et al., 2014: Do seasonal to decadal climate predictions underestimate the predictability of the real world? *Geophysical Research Letters*, 41, 5620–5628. doi:10.1002/2014GL061146

Smith, D.M. et al., 2019: Robust skill of decadal climate predictions. *Climate and Atmos. Sci.*, 13, doi: 10.1038/s41612-019-0071-y

Henley, B.J., 2017. Pacific decadal climate variability: indices, patterns and tropical-extratropical interactions. *Global and Planetary Change*, 155, 42-45. doi:10.1016/j.gloplacha.2017.06.004

Henley, B.J. et al., 2017: Spatial and temporal agreement in climate model simulations of the interdecadal pacific oscillation. *Env. Res. Lett.*, doi: <http://dx.doi.org/10.1088/1748-9326/aa5cc8>.

Folland, C.K. et al., 1999: Large scale modes of ocean surface temperature since the late nineteenth century. Chapter 4, pp73-102 of *Beyond El Nino: Decadal and Interdecadal Climate Variability*. Ed: A. Navarra. Springer-Verlag, Berlin, pp 374.

Chylek, P, et al., 2011: Ice-core data evidence for a prominent near 20 year time-scale of the Atlantic Multidecadal Oscillation. *Geophys. Res. Lett.*, 38, L13704, doi:10.1029/2011GL047501

Folland, C.K., et al., 2002: Relative influences of the Interdecadal Pacific Oscillation and ENSO on the South Pacific Convergence Zone. *Geophys. Res. Lett.*, 29, doi: 10.1029/2001GL014201. Pages 21-1 - 21-4.

Smith, D.M., et al, 2016: Role of volcanic and anthropogenic aerosols in the recent global surface warming slowdown. *Nat. Clim. Change*. 6, 936–940.

Remarks on Reviewer 1s comments

Summary Comments

Comment 1. Reviewer 1 was wrong but misled by the title of the paper. Moreover, some of the statements in the paper could give the wrong impression too if not read carefully.

Comment 2. Reviewer 1 was clearly wrong, but it would be better if the paper had some more explicit physical remarks in it. The strong arguments for anthropogenic effects don't come from statistics alone – they come from physical experiments and observations. This is covered in the quoted references as far I can see.

Comment 3. Reviewer 1 is clearly wrong. I have gone into this method in the past when collaborating with Mike Mann – it's a good statistical method..

Comment 4. Reviewer 1 s comment reflects the over interpretation of the control model results though I think it may also reflect his mis-understandings. However, whatever the reasons, I have some sympathy.

Comment 5. Reflects Reviewer 1's general unhappiness about the overall results which are indeed overstated, as well as some suspicion of MYM-SVD.

Comment 6. Rejecting the paper is certainly too strong, but it does need further work, and even then I don't think it is definitive enough for Nature Communications for the reasons I have given.

Major comments

Comment 1. Reviewer 1 is basically wrong. I agree with authors that the modern record is a mixture of natural and anthropogenic forcing, with lots of the latter in recent decades. I don't think further statistical analysis of the type Reviewer 1 wants is needed in this paper. What is needed are better anthropogenic aerosol forcings in particular, and model responses to them in the historical era where novel analytical methods might be used. The authors of course show the ensemble member clouds in their diagrams, so that comment is wrong.

Comment 2. Reviewer 1s comment is vague and seems to miss the main point. The authors give the correct reply within the limitations of their approach – which, as you can see, I think is overstated and in danger of being wrong in the longer term, so some unease is justified.

Comment 3. Reviewer 1 is wrong here.

Comment 4. I guess this will be acted on after publication of this paper somewhere.

Comment 5. These diagrams are now good.

Chris Folland

Reviewer #5 (Remarks to the Author):

Review of "Absence of internal multidecadal and interdecadal oscillations in the climate system" by Mann et al.

Recommendation: Major revisions

1) Throughout the manuscript the term "red noise" is used without defining it. I assume the authors mean by it the spectrum of an autoregressive process of first order AR(1). Its spectrum is typically called "red". However, it is Lorentzian in that its power increases initially but becomes white at lower-frequencies. A red noise process is a $1/f$ (f: frequency) process whose power increases continuously till zero frequency.

I would appreciate it if the authors could be more precise using this term. If they actually refer to the spectrum of an AR(1) process it would be helpful if the authors could also comment on the value of the AR(1) coefficient. That would help the readers (at least me) to understand the results much better.

2) Line 91: "warmth" is an unusual term. "heat" (if this is what you mean) would be more scientific and.

3) Lines 144-145: The study tries to detect AMO/AMV oscillations in a period range of 40-70 years in time series of 158 years. So that means you might be able to find 2-4 cycles in the data. So while for the PDO (15-20 year time scale) I find the results convincing, I am less convinced about the robustness of the results for the AMO. In that context I do not understand why the authors do not look at much longer control simulations (at least nowhere in the manuscript it is stated how long the used control simulation actually are).

4) Line 200: The statement here refers to an analysis CMIP5 simulations but that cites paper #54 which was published in 1991, well before CMIP5. This should be clarified.

5) Line 279: What is the meaning of "Wnk" and "lth"?

Figures 1 & 2: The caption should explain what figures a and b display. What are the units of the y-axis?

Responses to Review Comments

Comments indicated in *italics* and Responses in regular text.

Reviewers' Comments:

Reviewer #3 (Remarks to the Author):

My apologies to the authors for missing that the analysis was extended to a large multi-model ensemble rather the two examples used as illustrations.

In balance I like the scope of this article in applying a flexible spectral method to a large number of ensemble experiments. I feel that the significance testing is sound -- that is the statistical tests for spectral peaks hold their level. Thus this article confronts the community with a large number of control climate simulations that do not detect long term periodicities. "failing to reject the null hypothesis of no long term periodicities" Perhaps this finding justifies publication in Nature.

I note that this result comes with the focus on a single method: MTM-SVD. Given that the author is the developer of this method, it is natural for him to be a proponent. However, the issue of the power (in a statistical hypothesis testing context) of the MTM-SVD method when failing to reject a null hypothesis should be considered but given the authors response I suspect this is beyond the scope of their analysis and interests. A more positive approach to understand the power is to let the publication of this work to spur more scrutiny of the MTM-SVD method's (statistical) power and comparison with other approaches.

We appreciate the reviewer's positive overall assessment of our work. We agree that additional tests of the MTM-SVD method regarding its performance with respect to the detection of narrowband signals is beyond the scope of this article. We nonetheless would also like to emphasize that the MTM-SVD method has been subject to extensive and rigorous testing along these lines, as summarized in the article. Relevant specifically are the tests using synthetic data described in the Mann and Park (1999) review article. We have expanded the discussion in the revised manuscript to emphasize this point.

Reviewer #4 (Remarks to the Author):

General

As a new reviewer, it is clear that this paper has caused some confusion, and doubts about whether this paper is novel enough for Nature Communications. A basic problem is that the title does not accurately represent the content of the paper. Another problem for publication in Nature Communications is that current climate models, and certainly CMIP5 models run in control mode, are not good enough to make

definitive statements about the reality or otherwise of narrow band natural multidecadal and interdecadal variability. They are of relatively low resolution and it is very likely that much higher resolution is needed to capture longer time scale natural variability well. New, if preliminary, physical insights into fundamental model biases relating to resolution are in Scaife et al (2019), especially the last section of that paper. These insights are presented in the context of seasonal forecasts but have implications for all time scales and ocean-atmosphere interactions. Thus, it has subsequently been found that all current climate models suffer from the now well-known signal to noise paradox (Eade et al, 2014)) out to all time scales, also mentioned in Smith et al (2019) in the context of decadal prediction. This seems likely to especially affect the North Atlantic region and ocean-atmosphere interactions there. However, the results in this paper, with the modifications and additions suggested here, are certainly interesting enough for a more specialised publication, given acknowledgement of this potential problem but may be more marginal for Nature Communications. Having said that, if the somewhat wider perspectives suggested here were included, they would increase the suitability of the paper for Nature Communications. However, the analyses appear to need some modification as well, as mentioned below, to best make their points. Finally, the paper omits other references which cast significant light on how the topics in this paper should be approached, added below.

Having said this, there is nothing wrong as far as I can see with the statistical methods, though the paper does suffer a little from lacking an explicit dynamical background when discussing low frequency climate variability. As the authors clearly point out, it is clear that the observational record of the AMO and the PDO/IPO has been significantly affected by human-kind mainly via anthropogenic aerosols. The statistical approach of this paper may obscure this truth a little, but the quoted literature is clear, though the magnitudes of anthropogenic effects need more clarification in future. The authors in fact miss out an important reference about an anthropogenic influence on the PDO/IPO, listed below. This would strengthen the points already made about the observed record.

We have tremendous respect for the reviewer and thank him for his comments, which we recognize as being in good faith. While we differ with the reviewer, obviously, with regard to the significance and robustness of our findings, we have done our best to address the issues raised. We have accepted the reviewer's suggested change in title with minor modification. We have also cited the work mentioned by the reviewer and introduced appropriate caveats with regard to the faithfulness with which climate models reproduce internal dynamical variability. Please see our response to the reviewer's specific comments below.

Specific

1. I suggest that the title becomes "Absence of internal narrow band multidecadal and interdecadal oscillations in control simulations of the climate system".

We accept this suggestion with minor modification: “*Absence of internal narrow band multidecadal and interdecadal oscillations in climate model simulations*” which recognizes that our analysis of the forced simulations also does not support an internal (but rather external) mechanism.

2. Lines 23-33. A prominent c20 year PDO/IPO oscillation may be overstated as being current thinking. There is no settled theory of the PDO/IPO physics but there are a number of candidate processes which could give rise to several overlapping PDO/IPO time scales. These are summarised in the review by Henley (2017). This complexity tends to fit the analysis by Folland et al (1999) where their EOF3, which became the IPO of Power et al (1999), shows a vague power spectrum and a complex observational time series. Henley (2017) shows many spectra which support this vagueness as well as clear differences between the many analyses. Thus for the IPO, the reality of interdecadal to multidecadal variability might not depend on identifying a specific narrow band signal.

We have added a short discussion of this point referencing these studies.

3. Lines 34-56. This paper deliberately does not include paleoclimate analyses. However, this could be a serious weakness in the longer term for determining the truth about narrow band internal AMO oscillations. For instance, Chylek et al (2011) provide evidence for narrow band AMO oscillations around 20 and 50 years going back many centuries. If the paper, as modified, does not include paleo-analyses, this is all the more reason to be cautious about CMIP5 control model results.

While we do not analyze paleoclimate data in this study, we do allude to past studies using such data, e.g. last sentence of 4th paragraph “*Mann et al²¹ presented evidence based on analyses of paleoclimate proxy data that such a signal persists several centuries back in time.*” We also mention paleoclimate data in the final line of the article, “*Our findings also motivate a re-evaluation of the potential role of external natural radiative forcing in explaining evidence^{21,24} of low-frequency climate oscillations in paleoclimate proxy data, which should constitute the subject of future study.*” Here we are actually addressing the very point that the reviewer is making, albeit in a perhaps too terse manner. The point is that the paleoclimate data reflect both internal and external mechanisms, and this can be examined by looking at parallel MTM-SVD analyses of both the CMIP5 last millennium simulations and paleoclimate proxy data, the subject indeed of our ongoing work. We have added some additional language making this point at the end of the article.

4. Somewhere in the introductory section another caution should be expressed, given relatively short model or observational results. Wavelet analyses of the climate often show that specific narrow band oscillations are episodic. This is, for instance, seen in the Knight et al (2005) (quoted in the refs) 1400 year control model results and the Chylek et al (2011) paleoclimate results. So this adds an extra dimension to interpreting model or observational results. Episodes where the narrow band is missing can be a century, or more. long. This could affect ensemble member control

runs 150 years long where in some members a narrow band oscillation exists and in others of the same model it does not. This would not necessarily be an error, but it would make detection of narrow band oscillations in short control runs more challenging. Of course, this might also mean that the modern short observational record could overstate the long-term frequency or persistence of a specific apparent narrow band feature.

We've added some language to this effect in the process of responding to comment #2 above.

5. Line 140-143. It is not stated which surface temperature fields are being analysed and the captions miss this information. Are these global mean fields and time series? It would be better, given that the paper concentrates on the PDO/IPO and AMO, to use fields that best pick up these oscillations. These would be the North Atlantic region and an appropriate Pacific region. The need to do this is shown by the regional oceanic SST spectra of Folland et al (1999), which clearly differ a lot.

We have clarified in the figure captions, Methods section and main article that global gridded surface temperature fields were analyzed. We must differ with the reviewer on this point however. Seminal studies arguing for an AMO in both observations (Mann and Park, 1994) and model simulations (Knight et al, 2005) employed an MTM-SVD analysis of the global (rather than regional) surface temperature field. If a coherent large-scale narrowband signal exists, it will be detectable in the global surface temperature field. And only through a global analysis can we see the importance of teleconnections between basins etc. as argued for in a number of past studies. Moreover, a global analysis doesn't constrain the mechanisms to a particular ocean basin and so is more appropriate in the context of an exploratory analysis. While basin-restricted analyses are worthwhile, we do not think they are necessary in the context of the current study.

6. Lines 176 -230. An addition paragraph is needed. Having criticised CMIP5 models, there is a paper showing that most CMIP5 models reproduce the spatial pattern of the IPO well, though they do not perform so well on its temporal variations (Henley et al (2017). This suggests that there is genuine IPO physics distinguishing the IPO from ENSO, even if CMIP5 model temporal variability is poor. Furthermore, it suggests that the models pick up some of this physics. A "real" IPO (narrow band or not) is consistent with the observational paper of Folland et al (2002) showing a distinctive influence of the IPO compared to ENSO on the South Pacific Convergence Zone.

We've added some language to this effect in the introduction.

7. Lines 188-196 (probably). Add a mention of Smith (2016) which shows that anthropogenic aerosols have likely quite recently influenced the PDO.

Our feeling is that this is an overly specific point that doesn't seem relevant to the discussion since we find no evidence for a narrowband PDO-like signal in observations or models.

8. Lines 231-235. This part of the last paragraph of the main text is not sound. Not only does it overinterpret the results of this paper, but it contradicts the recent decadal prediction paper by Smith et al (2019) showing recent advances in prediction skill. Decadal prediction skill is clear over years 2-9 ahead in several variables, with additional predictability from predicted forcing changes. There clearly is skill in decadal forecasts this far ahead, particularly in the Atlantic sector. Thus Meehl et al (2014) (quoted in the refs) show generally useful CMIP5 decadal forecasting skill of the AMO out to at least years 3-6. Initial conditions themselves in decadal forecasts are of course influenced by anthropogenic effects, quite apart from skill due to forecast changes in forcing. So the control model results in this paper have few serious implications for forecasts of periods as short as a decade or somewhat less, the current time scales of decadal forecasts.

We have modified this paragraph as per our response to comment #3 above. With regard to the discussion specifically of Meehl et al and Smith et al 2019, we have clarified our point further. There is skill in initial value forecasts but as argued in ref 44, much of that skill seems to come from the specification of external forcing. E.g. recovery from post-volcanic cooling events in response to 1982 and 1991 eruptions. We have clarified this point in the revised m.s.

Minor comment

1. The caption to Fig 1 is wrong. It should read "control" on the first line. In Fig 1 and 2 captions, the a and b parts of the figures should be described under these letters.

Thanks--these have been fixed.

References

Scaife et al., 2019. Does increased atmospheric resolution improve seasonal climate predictions? Atmos Sci Lett., 20, doi: 10.1002/asl.922

Eade, R., et al., 2014: Do seasonal to decadal climate predictions underestimate the predictability of the real world? Geophysical Research Letters, 41, 5620–5628. doi:10.1002/2014GL061146

Smith, D.M. et al., 2019: Robust skill of decadal climate predictions. Climate and Atmos. Sci., 13, doi: 10.1038/s41612-019-0071-y

-Henley, B.J., 2017. Pacific decadal climate variability: indices, patterns and tropical-extratropical interactions. Global and Planetary Change, 155, 42-45. doi:10.1016/j.gloplacha.2017.06.004

Henley, B.J. et al., 2017: Spatial and temporal agreement in climate model simulations of the interdecadal pacific oscillation. Env. Res. Lett.,

doi: <http://dx.doi.org/10.1088/1748-9326/aa5cc8>.

-Folland, C.K.et al, 1999: Large scale modes of ocean surface temperature since the late nineteenth century. Chapter 4, pp73-102 of *Beyond El Nino: Decadal and Interdecadal Climate Variability*. Ed: A. Navarra. Springer-Verlag, Berlin, pp 374.

-Chylek, P, et al., 2011: Ice-core data evidence for a prominent near 20 year time-scale of the Atlantic Multidecadal Oscillation. *Geophys. Res. Lett.*, 38, L13704, doi:10.1029/2011GL047501

Folland, C.K., et al., 2002: Relative influences of the Interdecadal Pacific Oscillation and ENSO on the South Pacific Convergence Zone. *Geophys. Res. Lett.*, 29, doi: 10.1029/2001GL014201. Pages 21-1 - 21-4.

Smith, D.M., et al, 2016: Role of volcanic and anthropogenic aerosols in the recent global surface warming slowdown. *Nat. Clim. Change*. 6, 936–940.

We added the above references.

Remarks on Reviewer 1s comments

Summary Comments

Comment 1. Reviewer 1 was wrong but misled by the title of the paper. Moreover, some of the statements in the paper could give the wrong impression too if not read carefully.

Comment 2. Reviewer 1 was clearly wrong, but it would be better if the paper had some more explicit physical remarks in it. The strong arguments for anthropogenic effects don't come from statistics alone – they come from physical experiments and observations. This is covered in the quoted references as far I can see.

Comment 3. Reviewer 1 is clearly wrong. I have gone into this method in the past when collaborating with Mike Mann – it's a good statistical method..

Comment 4. Reviewer 1 s comment reflects the over interpretation of the control model results though I think it may also reflect his mis-understandings. However, whatever the reasons, I have some sympathy.

Comment 5. Reflects Reviewer 1's general unhappiness about the overall results which are indeed overstated, as well as some suspicion of MYM-SVD.

Comment 6. Rejecting the paper is certainly too strong, but it does need further work, and even then I don't think it is definitive enough for Nature Communications for the reasons I have given.

Major comments

Comment 1. Reviewer 1 is basically wrong. I agree with authors that the modern record is a mixture of natural and anthropogenic forcing, with lots of the latter in recent decades. I don't think further statistical analysis of the type Reviewer 1 wants is needed in this paper. What is needed are better anthropogenic aerosol forcings in particular, and model responses to them in the historical era where novel analytical methods might be used. The authors of course show the ensemble member clouds in their diagrams, so that comment is wrong.

Comment 2. Reviewer 1s comment is vague and seems to miss the main point. The authors give the correct reply within the limitations of their approach – which, as you can see, I think is overstated and in danger of being wrong in the longer term, so some unease is justified.

Comment 3. Reviewer 1 is wrong here.

Comment 4. I guess this will be acted on after publication of this paper somewhere.

Comment 5. These diagrams are now good.

Chris Folland

We appreciate the reviewer #4 (Dr. Folland) assessment of the original reviewer #1 comments and are essentially in agreement with him. We believe we have addressed any substantive new issues raised by reviewer #4 in our responses above.

Reviewer #5 (Remarks to the Author):

Recommendation: Major revisions

1) Throughout the manuscript the term “red noise” is used without defining it. I assume the authors mean by it the spectrum of an autoregressive process of first order AR(1). Its spectrum is typically called “red”. However, it is Lorentzian in that its power increases initially but becomes white at lower-frequencies. A red noise process is a $1/f$ (f : frequency) process whose power increases continuously till zero frequency.

I would appreciate it if the authors could be more precise using this term. If they actually refer to the spectrum of an AR(1) process it would be helpful if the authors could also comment on the value of the AR(1) coefficient. That would help the readers (at least me) to understand the results much better.

The comment is fair enough. The term “red noise” in and of itself is subject to differing usage and definitions. In the climate time series analysis literature “red noise” is typically understood to be AR(1) red noise. We have revised the m.s. to clarify that this is indeed what we mean in our usage of the term.

2) *Line 91: “warmth” is an unusual term. “heat” (if this is what you mean) would be more scientific and.*

We have made the suggested change.

3) *Lines 144-145: The study tries to detect AMO/AMV oscillations in a period range of 40-70 years in time series of 158 years. So that means you might be able to find 2-4 cycles in the data. So while for the PDO (15-20 year time scale) I find the results convincing, I am less convinced about the robustness of the results for the AMO. In that context I do not understand why the authors do not look at much longer control simulations (at least nowhere in the manuscript it is stated how long the used control simulation actually are).*

It is important to note that the burden of proof is on demonstrating that a significant oscillation exists (i.e. rejecting the null hypothesis), and to the extent that it is difficult to do so with short times series, that is of course a major caveat with respect to the claim that such oscillations exists! But we appreciate the larger point being made by the reviewer.

We note that our analysis of CMIP5 control simulations includes a substantial fraction ($N=21$, just under half the total $N=44$) that are 500 years or longer, i.e. the length of roughly ten putative multidecadal cycles. Figure 4c actually shows one such case (one of the few models that does exhibit a multidecadal oscillation as shown, illustrating more than ten cycles of the oscillation). So we would argue that our analysis of the CMIP5 control simulations is more than adequate to assess whether a statistically significant multidecadal oscillation exists. We have added language to this effect in the revised m.s.

We have also revised the tables in the Supplementary Information to specify the length of the simulations.

4) *Line 200: The statement here refers to an analysis CMIP5 simulations but that cites paper #54 which was published in 1991, well before CMIP5. This should be clarified.*

There was an error in the sequencing of the references. We have fixed it (the correct reference is Stocker et al IPCC 2013 chapter).

5) *Line 279: What is the meaning of “Wnk” and “lth”?*

There were multiple problems there. “k” should be “l” and there were missing subscripts and superscripts and italicization. These have now been fixed.

Figures 1 & 2: The caption should explain what figures a and b display. What are the units of the y-axis?

The *a* and *b* labels were erroneous. What was labeled “a” is simply the inset that is referred to already in the figure caption. We simply removed the “a” and “b” labels.

REVIEWERS' COMMENTS:

Reviewer #4 (Remarks to the Author):

1. The authors have responded to my comments well on the whole, and the paper is much better. I still remain cautious, but the paper will be useful in challenging the climate community. Thus as far as it goes, the paper seems sound but time will tell whether these results will stand up given model limitations, now better referenced. So I recommend publication in Nature Communications.

2. I have one substantive comment. Although the authors are vague on this point, the lack of narrow band modes of natural interdecadal variability does not prove that physically based interdecadal modes of variability like the IPO or AMO do not exist. I don't think the authors mean to imply this, but they could be misunderstood. Thus the quasi-continuous spectrum of the IPO reveals this lack of narrow band behaviour, as the authors now discuss. This is unsurprising given the several possible internal physical processes that have been suggested to exist on decadal and longer time scales in the Pacific and through interactions with other oceans. These influences could easily greatly vary in relative importance, including stochastic forcing, making the IPO mode difficult to predict and a challenge for decadal prediction – as the authors imply. The same might be true for the AMO.

To avoid doubt, the authors should add a sentence or two of clarification in their concluding comments.

Chris Folland